# Critical Role of Neuronal Vps35 in Blood Vessel Branching and Maturation in Developing Mouse Brain

**DOI:** 10.3390/biomedicines10071653

**Published:** 2022-07-09

**Authors:** Yang Zhao, Daehoon Lee, Xiao-Juan Zhu, Wen-Cheng Xiong

**Affiliations:** 1Key Laboratory of Molecular Epigenetics of Ministry of Education, Institute of Cytology and Genetics, Northeast Normal University, Changchun 130024, China; zhaoy148@nenu.edu.cn; 2Department of Neurosciences, School of Medicine, Case Western Reserve University, Cleveland, OH 44106, USA; dxl660@case.edu

**Keywords:** Vps35, blood vessel, astrocyte, microglia

## Abstract

Vps35 (vacuolar protein sorting 35), a key component of retromer, plays a crucial role in selective retrieval of transmembrane proteins from endosomes to trans-Golgi networks. Dysfunctional Vps35/retromer is a risk factor for the development of neurodegenerative diseases. Vps35 is highly expressed in developing pyramidal neurons, both in the mouse neocortex and hippocampus, Although embryonic neuronal Vps35’s function in promoting neuronal terminal differentiation and survival is evident, it remains unclear whether and how neuronal Vps35 communicates with other types of brain cells, such as blood vessels (BVs), which are essential for supplying nutrients to neurons. Dysfunctional BVs contribute to the pathogenesis of various neurodegenerative disorders. Here, we provide evidence for embryonic neuronal Vps35 as critical for BV branching and maturation in the developing mouse brain. Selectively knocking out (KO) Vps35 in mouse embryonic, not postnatal, neurons results in reductions in BV branching and density, arteriole diameter, and BV-associated pericytes and microglia but an increase in BV-associated reactive astrocytes. Deletion of microglia by PLX3397 enhances these BV deficits in mutant mice. These results reveal the function of neuronal Vps35 in neurovascular coupling in the developing mouse brain and implicate BV-associated microglia as underlying this event.

## 1. Introduction

During mouse embryogenesis, vessel formation occurs via two mechanisms: vasculogenesis and angiogenesis. At embryonic 8.5 (E8.5), vascular EC precursors coalesce to form a primitive vascular plexus—a process termed vasculogenesis. Later, this primary vascular network gives rise to the arteries and veins of the leptomeninges (pia mater and arachnoidea) [14,15]. Subsequently, at E9.5, EC sprouts from pre-existing vessels to extend the vascular plexus—a process termed angiogenesis [16,17,18]. Finally, branches anastomose with other branches, forming a rich capillary plexus. During postnatal development (P4/P8), brain angiogenesis is highly dynamic, and sprouting occurs throughout different regions, such as the cortex and hippocampus, and EC tip cells grow in various directions. CNS vascular remodeling is completed around P24 [15,19].

The brain consumes the most energy of all organs in the body, although it comprises only 2% of body weight [1,2]. However, the brain lacks a reservoir to store fuel for use when needed; this high energy demand is supplied by an extensive cerebral vasculature, which can ensure appropriate supply of energy substrates, oxygen, and glucose for neurons [3,4]. The cerebral blood vessels (BVs) consist of interconnected arteries, capillaries, and veins, which are all made up of endothelial cells (ECs). Newly formed vessels are stabilized by the recruitment of supporting cells. Whereas vascular smooth muscle cells (VSMCs) are supporting cells for arteries, pericytes support capillaries and veins [5,6]. In addition to providing nutritional support, BVs in the CNS have unique properties that allow them to form the blood–brain barrier (BBB) [7]. The BBB is created by tight junctions (TJs) between the endothelial cells (ECs) of the capillaries that perfuse the brain parenchyma, which is a key feature of the BBB and significantly reduces permeation of polar solutes through paracellular diffusional pathways between the ECs from the blood plasma to the brain extracellular fluid [8,9]. The BBB is a multicellular structure. The abluminal surface of the endothelium is surrounded by a basal lamina, pericytes and astrocyte end-feet and may also interact directly or indirectly with neurons [7]. In addition, microglia have been found to interact with CNS vessels and regulate immune cell passage across the BBB [10,11]. Proper neuronal function depends on precise ionic concentrations in brain parenchyma, which are provided by the highly regulated BBB. Furthermore, the BBB also can limit the entry of toxins, pathogens, and immune cells into neural tissue during injury and disease [10]. Dysfunction of the BBB in mice is implicated in multiple neurological diseases, such as Alzheimer’s disease (AD) [12,13].

It has been found that the vascular network closely associates with the neuronal network throughout development. During CNS vascularization, neurons regulate CNS-specific angiogenesis through classical angiogenic factors, such as VEGF, Nogo-A, and Slit2. In addition, neuronal activity is important for controlling cerebral blood flow, and a reduction in the sensory input was found to lead to decreased EC proliferation, vascular density, and branching in the barrel cortex [19,20,21,22]. For example, neuronal activity leads to a rise in astrocytes [Ca^2+^]_i_, which can activate large conductance Ca^2+^-activated K^+^ channels on astrocyte end-feet, leading to an efflux of K^+^ onto the vasculature and dilation of arterioles [23]. The coupling between neuronal activity and cerebral blood flow is termed functional hyperemia or neurovascular coupling (NVC) [24]. Pericytes envelop the surface of the vascular tube, serving not only as scaffolding but also communicating with ECs by direct physical contact and paracrine signaling pathways [25,26]. Studies of pericytes have suggested numerous vascular functions, including regulation of cerebral blood flow, maintenance of the BBB, and control of vascular development and angiogenesis. Additionally, pericyte degeneration and loss have been demonstrated in AD patients and animal models [27,28,29,30,31]. Similarly, astrocytes, a main glial subset, are known as a type of BV-supporting cells. Astrocytes have been found to play a role in postnatal angiogenesis [32,33], and they contribute to the formation of the BBB and maintenance of BBB integrity, likely by communicating with pericytes [34,35]. They also bridge neuronal activity to regulate local blood flow with fine processes and end-feet that intimately contact both neuronal synapses and the cerebral vasculature [36,37,38]. Microglia, residents of macrophages in the brain, also regulate brain angiogenesis, and they are often associated with newly formed BVs, regulating BV branching and homeostasis [39,40,41].

The concept of the neurovascular unit (NVU) was introduced to emphasize the unique relationship between brain cells and cerebral vasculature [4]. The NVU consists of brain parenchymal cells, including excitatory neurons, inhibitory interneurons, astrocytes, and microglia interacting with vascular cells, including pericytes, VSMCs, and ECs [42]. The NVU plays important roles in maintaining brain function, particularly the regulation of cerebral blood flow (CBF) and formation of the BBB [3,4,27,36,37]. In addition, alterations of the NVU are anticipated to lead to brain dysfunction and damage and involve in a wide variety of neurovascular alterations in neurodegenerative diseases [43,44]. Thus, it is necessary to investigate the effects of NVU on BV development and homeostasis.

Vacuolar protein sorting 35 (Vps35) is a key component of retromer that selectively sorts transmembrane proteins/cargos to the trans-Golgi network or plasma membrane [45,46,47]. Deficiency of Vps35 has been reported as an active player in the pathogenesis of neurodegenerative disease [48,49,50]. Vps35 is highly expressed in developing pyramidal neurons. We previous showed that Vps35-loss in mouse embryonic pyramidal neurons results in not only terminal differentiation deficit and maturation defects but also neurodegenerative pathology, such as glial pathology [51]. However, Vps35’s role in developing pyramidal neurons during BV development and function remains unexplored.

Here, we provide evidence of the function of developing pyramidal neuron Vps35 in regulating BV branching and maturation in the developing mouse cortex and hippocampus. Developing pyramidal neuron-specific Vps35 KO cortex and hippocampus presented with fewer veins and capillaries, smaller arterioles, fewer pericytes, and impaired vascular basement membranes (vBMs). Additionally, these vessel phenotypes were associated with an increase in BV-associated reactive astrocytes and a reduction in BV-associated microglia. After deletion of microglia by PLX3397, the BV deficits in mutant mice were enhanced, and the BV-associated reactive astrocytes had exhibited more drastic enhancement. In aggregates, these results reveal the role of pyramidal neuron Vps35 in regulating BV development by regulating the NVU, and BV-associated microglia may have an important function.

## 2. Materials and Methods

### 2.1. Animals

Mice were cared for according to animal protocols approved by Case Western Reserve University (CWRU) according to the National Institute of Health (NIH) guidelines. All mice were housed in standard conditions with food and water provided and a 12 h dark/light cycle. Vps35-floxed (*Vps35^f/f^*) mice were generated, maintained, and genotyped as described previously [52,53]. *GFAP-Cre* mice (stock 004600), as well as *Emx1-Cre* (stock 005628) and *Camk2a-Cre* (stock 027310) mice, were purchased from Jackson Laboratories. *NeuroD6-Cre* (also called Nex-Cre) mice were kindly provided by Klaus-Armin Nave. The *Vps35^f/f^* mouse line was crossed with above *Cre* mouse lines to generate Vps35 homozygous mutant *Vps35^Neurod6^*, *Vps35^GFAP^*, *Vps35^Emx1^*, and *Vps35^Camk2a^* mice. All of the mouse lines indicated above were maintained in a C57BL/6 background for more than six generations. Mice were housed in 12/12 h light/dark cycle animal rooms. Both male and female mice were examined throughout all the experiments.

### 2.2. Antibodies and Reagents

The following antibodies were used: rabbit polyclonal anti-Vps35 antibody was generated against the murine Vps35 C-terminal sequence as described previously [51] (diluted 1:1000); mouse monoclonal anti-β-Actin (diluted 1:1000; Cell Signaling, Danvers, MA, USA, 3700S); rabbit polyclonal anti-Vps26a (diluted 1:500; abcam, Cambridge, UK, 23892); goat polyclonal Iba1 (diluted 1:500; abcam, ab5076); rabbit monoclonal anti-GFAP (diluted 1:200; Cell Signaling, 12389); rabbit polyclonal anti-cleaved caspase-3 (diluted 1:300; Cell Signaling, 9661); mouse monoclonal anti-Vps29 (diluted 1:200; Santa Cruz, Dallas, TX, USA, sc-398874); rat monoclonal anti-CD31 (diluted 1:200; BD, Franklin Lakes, NJ, USA, 553369); rabbit polyclonal anti-SLC16A1 (diluted 1:200; Origene, Rockville, MD, USA, TA321556); rat monoclonal anti-PDGFRβ (diluted 1:200; Cell signaling technology, 3169); monoclonal anti-SMA-647 (diluted 1:1000; Santa Cruz Biotechnology, sc-32251); mouse monoclonal anti-laminin γ1 (diluted 1:500; DSHB, Iowa City, IA, USA, 2E8); and fluorescent-labeled secondary antibodies (diluted 1:1000; Thermo Fisher Scientific, Waltham, MA, USA, Alexa Fluor conjugates); HRP-labeled antibodies (diluted 1:1000; Invitrogen, Waltham, MA, USA).

The following reagents were used: DAPI (Thermo Fisher Scientific), EdU (Sigma, St. Louis, MO, USA), PLX3397 (MedChemExpress, Princeton, NJ, USA), and protease inhibitor cocktail (Roche).

### 2.3. Tissue Processing and Immunofluorescence

Mice with indicated genotypes were anesthetized and perfused transcardially with 4% paraformaldehyde (PFA, Germantown, WI, USA; pH 7.4). The brains were dissociated and postfixed overnight, then mounted in agarose prior to vibratome sectioning. Next, 35–50 µm free-floating vibratome sections of brains were collected in PBS, then blocked and permeabilized with blocking solution containing 10% BSA and 0.5% Triton X-100 in PBS for 1 h at room temperature and incubated with primary antibodies in blocking solution overnight at 4 °C. The brain sections were then washed with 3× PBST buffer, then incubated with appropriate secondary antibodies in a ratio of 1:1000 (Thermo Fisher Scientific, Alexa Fluor conjugates) for 2 h at room temperature. Brain sections were washed and incubated with DAPI (Thermo Fisher Scientific) to reveal cell nuclei. After washing with PBS, mouse brain sections were mounted for confocal imaging with a Zeiss LSM 800 system. Images were also captured with a BZX fluorescence microscope.

### 2.4. Western Blotting

Tissues were dissected and homogenized in lysis buffer (50 mM Tris-HCl (Ph 7.4), 150 mM NaCl, 1% TritonX-100, 0.1% SDS, 10% Glycerol, and 1 mM NaF) and supplemented with protease inhibitor cocktail (Roche). After being incubated on ice for 20 min frozen and thawed twice, and ultracentrifuged at 10,000× *g* at 4 °C for 30 min, the protein supernatant was obtained. Protein concentration was measured by BCA assay (Pierce Biotechnology, Waltham, MA, USA). The supernatants were mixed with loading buffer for Western blot analysis. Based on the protein molecular weight, we used 10% SDS-PAGE gels to separate proteins and then transferred them onto nitrocellulose blotting membranes. After blocking with 5% low-fat dried milk (in 1× TBST) for 1 h at room temperature, antigen-specific primary antibodies were diluted to incubate the membrane overnight at 4 °C; then, species-specific horseradish-peroxidase-conjugated secondary antibodies (1:5000, Thermo) and an ECL kit (Pierce, Rockford, IL, USA) were used to visualize the target proteins. Image J software was used to analyze the protein bands, which were normalized to loading control (β-actin).

### 2.5. Quantitative Real-Time RT-PCR (qRT-PCR)

Total RNA was extracted from the cortex and hippocampus using TRIzol reagent (Life Technologies, Carlsbad, CA, USA), as previously described [54]. Then, purified RNA (5 µg) was used for cDNA synthesis with a GoScript reverse transcription system (Promega, Madison, WI, USA). cDNA products were subjected to subsequent qPCR using SYBR green (QIAGEN, Hilden, Germany) in a CFX96 real-time system (Bio-Rad, Hercules, CA, USA). Each sample was repeated at least 3 times, and the mRNA level was normalized to GAPDH using the 2^−ΔΔCt^ method. Primers for individual genes were as follows: GFAP (forward: 5′-CACCT ACAGG AAATT GCTGG AGG-3′, reverse: 5′-CCACG ATGTT CCTCT TGAGG TG-3′); Vps35 (forward: 5′-GACTT CGCTG ATGAA CAGAG CC-3′, reverse: 5′-TCAGC GGATT CGCTT CACAC TG-3′); VEGFa (forward: 5′-GCAGG CTGCT GTAAC GATGA-3′, reverse: 5′-TCATG CGGAT CAAAC CTCAC-3′); VEGFb (forward: 5′-ACTGG GCAAC ACCAA GTCCG AA-3′, reverse: 5′-CCTGG AAGAA CACAG CCAAT GTG-3′); VEGFc (forward: 5′-CCTGA ATCCT GGGAA ATGTG CC-3′, reverse: 5′-ACAGA AGACC GTGTG CGAAT CG-3′); PDGFa (forward: 5′-CTGGC TCGAA GTCAG ATCCA CA-3′; reverse: 5′-GAGGA TGCCT TGGAG ACAAG TC-3′); PDGFb (forward: 5′-AATGC TGAGC GACCA CTCCA TC-3′, reverse: 5′-GCTGG ACTTG AACAT GACCC GA-3′); netrin-1 (forward: 5′-GTCTG GTGTG TGACT GTAGG CA-3′, reverse: 5′-CAACT GCAAC CTCCA TGCTC GG-3′); netrin-3 (forward: 5′-AGGTT TCCAG CAGAG CCGTT CT-3′, reverse: 5′-CTGTG AGTCA CATTG CAGAC CTG-3′); netrin-4 (forward: 5′-TGACC AGTGC TTACC TGTGG AG-3′, reverse: 5′-CACAA CACAG CAGGC AGCCA CT-3′); and GAPDH (forward: 5′-CATCA CTGCC ACCCA GAAGA CTG-3′, reverse: 5′-ATGCC AGTGA GCTTC CCGTT CAG-3′).

### 2.6. Statistics

All data are presented as mean ± SEM. GraphPad Prism 6 (GraphPad Software, San Diego, CA, USA) was used for statistical analysis. A two-tailed unpaired *t*-test was used to evaluate the statistical significance of two groups of samples. One-way analysis of variance (ANOVA) with a Tukey post hoc test was used to evaluate statistical significance of three or more groups of samples. The n numbers can be found in the figure legends. Statistical significance was defined as *p* < 0.05.

## 3. Results

### 3.1. Pyramidal Neuronal Vps35-Promoting PECAM-1^+^ BV BRANCHING and Morphogenesis in Developing Mouse Cortex and Hippocampus

To investigate Vps35’s function in blood vessel development in the brain, we generated several Vps35 cKO mouse lines, named *Vps35^GFAP^*, *Vps35^Emx1^*, and *Vps35^Neurod6^*, by crossing the floxed Vps35 allele (*Vps35^f/f^*) with *hGFAP-Cre*, *Emx1-Cre*, and *Neurod6-Cre* mice, respectively. The *hGFAP-Cre* mouse line expresses Cre not only in astrocytes but also in radial glial cells (RGCs), neural stem cells (NSCs), and their progenies beginning at E13.5 [55] (Appendix A). The *Emx1-Cre* expresses Cre in progenitors and postmitotic neurons of telencephalic cortical divisions, including RGCs, Cajal–Retzius cells, glutamatergic neurons, and astrocytes beginning at E10.5 [56] (Appendix A). *Neurod6-Cre* expresses Cre selectively in pyramidal neurons in the neocortex and granule and mossy cells of dentate gyri in the hippocampus starting at E11.5 [57]. Western blot analysis showed reductions in Vps35 protein levels in the cortex and hippocampus of the three Vps35 cKO mouse lines (Appendix A and Figure 1A–C). Vascular density and branching progressively increased after birth, reaching a peak at postnatal day 14 [22,58]. We thus first examined BVs in the three Vps35 cKO brains at an age of P14. Immune-staining analysis using antibodies against PECAM-1, a marker of BV-ECs, showed less PECAM-1^+^ vessels in the cortex and hippocampus of *Vps35^GFAP^* mice but not other brain regions (Appendix A). Quantification analyses indicated significant reductions in the total length and branch points of PECAM-1^+^ vessels in *Vps35^GFAP^* cortex and hippocampus (Appendix A). These results suggest a role of Vps35 in GFAP-Cre^+^ cells, including astrocytes or RGCs and neurons, in regulating cortical BV morphogenesis. Interestingly, similar BV deficits, including reduced BV total length and decreased BV branching, were also detected in *Vps35^Emx1^* mice (Appendix A), as well as *Vps35^Neurod6^* mice (Figure 1), in both the cortex and hippocampus (Appendix A, Figure 1D–I). Carefully comparing BV phenotypes among the three Vps35 cKO brains indicated a more dramatic deficit in BV branching in *Vps35^Neurod6^* mice (Figure 2). These results suggest that the neuronal Vps35 is necessary for BV development, in particular, BV branching, in the P14 developing brain.

### 3.2. P14 to P21, a Critical Period for Neuronal Vps35 Regulation of BV Morphogenesis

We next asked whether neuronal Vps35 regulation of BV morphogenesis is age-dependent. In addition to *Vps35^Neurod6^* mice, we generated Vps35; Camk2a-Cre mice, named as *Vps35^Camk2a^*, by crossing *Vps35^f/f^* mice with *Camk2a-Cre* mice, which induce recombination in postmitotic cortical and hippocampal pyramidal neurons and are absent from neural precursors and glial cells after P1 [59] (Appendix A). In contrast to *Vps35^Neurod6^* mice, which have neonatal lethality before P23 [51], *Vps35^Camk2a^* mice had a relatively normal life span (data not shown). Interestingly, imaging analysis of PECAM-1^+^ BVs showed little to no obvious change in BVs between control and *Vps35^Camk2a^* brains at an age of P60 (Appendix A), indicating a minimal role of Vps35 in Camk2a-Cre^+^ neurons in regulating BV morphogenesis. Together, these results suggest that the expression of pyramidal neuronal Vps35 in the developing (E13.5 to ~P14) brain is necessary for BV morphogenesis, implicating a critical period (e.g., E13.5 to P14) for this event.

We further tested this view by examining cortical brain BV morphology (e.g., length and branches) at various neonatal ages (e.g., P7, P14, and P21) in control and *Vps35^Neurod6^* mice. The BV morphology was observed by coimmunostaining analysis using antibodies against PECAM-1 (a marker for ECs in all types of BVs) and SLC16A1 (solute carrier family 16 member 1) (a marker of ECs of veins and capillaries) [60,61]. At P7, the BV length and branches in the mutant cortex were comparable with those of control mice (Figure 3A,C–E), but at both P14 and P21, decreased BV densities and branches became obvious in the mutant mice (Figure 3A,C–E). These results further defined the critical time window, P14–P21, for BV deficits in *Vps35^Neurod6^* mice, implicating a critical role of neuronal Vps35 in promoting BV branching/maturation. These results also suggest that a decrease BV density and branch points in the *Vps35^Neurod6^* cortex are likely occur largely in veins and capillaries.

### 3.3. Decreased Pericytes and Thinner Arterioles in Vps35^Neurod6^ Cortex

We then examined other types of BV-associated cells, such as pericytes and arterioles. in the cortex of control and *Vps35^Neurod6^* mice. Pericytes, marked by PDGFRβ, wrap around blood capillaries/veins not only as scaffolding but also communicating with ECs by direct physical contact and paracrine signaling pathways [29,62]. As shown in Figure 4A–C, PDGFRβ^+^ pericytes were tightly associated with SLC16A1^+^ capillaries/veins, which were markedly reduced in the mutant cortex at P14 and P21 but not P7, as compared with those of littermate controls (Figure 4A,B), suggesting an age-dependent BV-pericyte deficit. Notice that there were PDGFRβ^+^, but SLC16A1^−^ BVs in P14 and P21 control mice, and were undetectable in the *Vps35^Neurod6^* mice (Figure 4A,C). Considering reports that PDGFRβ is also expressed in smooth muscle cells (SMCs), in addition to pericytes [63,64], these results suggest that both pericytes and SMCs might be reduced in *Vps35^Neurod6^* mutant mice. We then asked whether SMA (smooth muscle actin)-marked arterioles were deficient in the mutant mice. Immunostaining analysis using anti-SMA showed obviously thinner arterioles in the mutant cortex than those of controls, whereas the length of arterioles in the mutant cortex appeared to be comparable with that of control mice (Figure 4D–G). These results thus demonstrate deficits in BV veins and capillaries, as well as arterioles, in P14 and P21 *Vps35^Neurod6^* mutant mice.

### 3.4. Elevated Apoptosis in ECs but Not Pericytes in Vps35^Neurod6^ Cortex

To understand how the reduced BV length and branch points are induced in the *Vps35^Neurod6^* cortex, we speculate that the BV ECs and BV pericytes in the mutant mice may have reduced proliferation and/or increased cell death—two critical processes for BV homeostasis. To test this view, control and *Vps35^Neurod6^* mutant mice (at P14) were injected with EdU (a marker of cell proliferation) 24 h before their sacrifice (Figure 5A) [65]. Immunostaining analysis showed that there were substantial EdU^+^ cells in the mutant cortex, without an obvious difference from that of control mice (Figure 5B). Further analysis demonstrates that a small fraction of EdU^+^ cells was coimmunostained with PECAM-1^+^ ECs in both the control and mutant cortex (Figure 5B,C), and the percentage of EdU ^+^ PECAM-1^+^ cells relative to the total EdU^+^ cells was comparable between control and mutant mice (Figure 5B,C). These results thus eliminate the possibility of EC proliferation deficit causing BV reduction. We then examined active/cleaved Caspase 3^+^ apoptosis in ECs and pericytes in the control and mutant cortex. Interestingly, an increase in cleaved-Caspase 3^+^ PECAM-1^+^ ECs was detected in *Vps35^Neurod6^* mutant mice (Figure 5D,E). However, few to no cleaved Caspase 3^+^ PDGFRβ^+^ pericytes were detectable (Figure 5F,G). These results suggest that increased apoptotic ECs may be one of the cellular mechanisms of reduced BV density in mutant mice.

### 3.5. Increased BV-Associated Reactive Astrocytes in Vps35^Neurod6^ Brain

An NVU (neurovascular unit) consists of neurons, astrocytes, microglia, pericytes, VSMCs (vascular smooth muscle cells), and ECs. We thus asked whether astrocytes are altered in the mutant cortex, as they fulfill critical functions in regulating BV homeostasis and function [36,42]. Coimmunostaining analysis using antibodies against SMA, PECAM-1, and GFAP (a marker for reactive astrocytes) showed that in P14 and P21 cortexes of control mice, a small number of GFAP^+^ astrocytes were detectable, which were tightly associated with the SMA^+^ large vessels or arterioles (Figure 6A,C). However, in Vps35 mutant mice, the GFAP^+^ astrocytes were significantly increased (Figure 6A–D). In addition to the large vessels/arterioles, the GFAP^+^ astrocytes were also associated with the small vessels (veins and capillaries), as these vessels were marked with both PECAM-1 and SLC16A1 (Figure 6A–D). The increased GFAP protein level in *Vps35^Neurod6^* cortical homogenates at P14 was confirmed by Western blot analyses (Figure 6E,F). We also detected increased *GFAP* mRNA level by RT-PCR in *Vps35^Neurod6^* mice at P21 (Figure 6G). These results suggest a marked increase in BV-associated GFAP^+^ reactive astrocytes in the mutant cortex. Additionally, examining GFAP^+^ astrocytes in the hippocampus showed a similar increase in BV-associated GFAP^+^ astrocytes (Appendix A). Again, at P7, few to no changes in SMA^+^ arterioles or BV-associated GFAP^+^ astrocytes were detected in the mutant mice (Appendix A).

### 3.6. Increased Microglial Activation but Decreased BV-Associated Microglia in Vps35^Neurod6^ Cortex

We then asked whether microglia, in particular, BV-associated microglia, are altered in the mutant cortex, as microglia are another cellular component of the NVU [40]. Coimmunostaining analysis using antibodies against Iba1 (a marker for microglia) and PECAM-1^+^ ECs showed elevated Iba1^+^ microglial cell intensity in the mutant cortex (Figure 7A–D), in line with our previous reports [51,66]. However, in contrast to astrocytes, BV-associated Iba1^+^ microglia were decreased compared with control mice (Figure 7A–D). These results implicate microglial cells in regulation of BV morphogenesis in the developing cortex.

### 3.7. BV-Associated Microglia in Promoting BV Branching and Morphogenesis

We further tested the function of microglia in BV morphogenesis in the developing cortex by injecting PLX3397 into control and *Vps35^Neurod6^* pups to deplete microglial cells, as illustrated in Figure 8A. PLX3397 is an antagonist of colony-stimulating factor 1 receptor (CSF1R), which is often used to block microglial proliferation and survival, depleting microglia [67,68]. As shown in Figure 8B, upon PLX3397 treatments, Iba1^+^ microglia were largely depleted in the cortex of both control and *Vps35^Neurod6^* mutant mice (Figure 7B,C). Few Iba1^+^ cells remained in the control and mutant cortex after PLX3397 treatments, exhibiting larger cell soma and shorter processes (Figure 8B,D). These morphologically distinct Iba1^+^ cells might be recovered myeloid cells [69,70,71,72]. Unexpectedly, the BV density, in particular, the BV branch points, were further reduced in *Vps35^Neurod6^* mutant mice treated with PLX3397 as compared with those of Veh-treated mutant mice (Figure 8B,F,G). The reductions in BV density and branch points by PLX3397 were also detectable in the control mice (Figure 8B,F,G). Simultaneously, we examined the GFAP^+^ astrocytes in the mice treated with Veh or PLX3397. Interestingly, a more dramatic increase in GFAP^+^ astrocytes was detected in the PLX3397-treated mutant mice than that in mutant mice treated with DMSO (Figure 8B,E,H). In control mice, the level of active GFAP^+^ astrocytes did not significantly change after treatment with PLX3397 (Figure 8B,E). Overall, these results suggest that microglia, in particular, BV-associated microglia, may modulate the status of astrocytes to affect BV development.

### 3.8. Little Change in Expression of Genes Involved in Angiogenesis in Vps35^Neurod6^ Cortex and Hippocampus

To understand molecular mechanisms underlying EC death and decreased BV branches and length in the mutant mice, we examined the expression of genes known to be critical for angiogenesis and EC survival, such as VEGFs (vascular endothelial growth factors), PDGFs (platelet-derived growth factors), and netrins [73,74,75,76,77]. However, real-time PCR analysis showed little to no change in the expression of these genes in the mutant cortex or hippocampus, as compared to those of controls (Appendix A). These results suggest that the reduction in BV branching/density and the increase in EC apoptosis may be induced by VEGF/PDGF/netrin-independent mechanisms. Alternatively, it is possible that there is a small change in cellular VEGFs/PDGFs/netrins that was undetectable by RT-PCR analysis of their transcripts in the whole cortex.

## 4. Discussion

Vps35 dysfunction is known to be a risk factor for the development of neurodegenerative diseases [51,78]. Vps35 is highly expressed in the developing pyramidal neurons of the mouse neocortex and hippocampus [51]. However, the function of neuronal Vps35′ in neurovascular communication remains elusive. Here, we provide evidence that embryonic neuronal Vps35 is critical for BV branching and maturation in the developing mouse brain. We found that *Vps35* deficiency in developing neurons (not postnatal) results in reduced BV branching and density (Figure 1), arteriole diameter (Figure 4), and BV-associated pericytes and microglia (Figure 4 and Figure 7) but an increase in BV-associated reactive astrocytes (Figure 6 and Appendix A). Further analysis showed that deletion of microglia by PLX3397 enhances BV deficits in Vps35 mutant mice (Figure 8). These results reveal the function of neuronal Vps35 in regulating the neurovascular unit in the developing mouse brain.

Previous research on the function of embryonic neuronal Vps35 suggested a critical role of Vps35 in neuronal terminal differentiation and survival [51]. In our studies, we found that neuronal Vps35 is also involved in regulating BV development and maturation. These functions with respect to regulation of BV morphogenesis are age-dependent, as we did not detect an obvious change in BVs in *Vps35^Camk2a^* brains compared with control mice (Appendix A). In addition, we found that BVs deficits occur in a critical time window, i.e., P14–P21 (Figure 3), suggesting that the function of neuronal Vps35 in promoting BV branching/maturation.

How does Vps35 regulate BV branching/maturation? The mechanisms underlying BV deficiency remains unclear. Previous research showed that Vps35 deficiency impairs neuronal terminal differentiation and survival [51]. During embryonic and early postnatal development, neurons play a fundamental role in CNS vascularization, and both neuronal signal and function have an effect on vasculature formation [22,79,80,81,82]. In addition, neurons either signal directly to blood vessels or activate astrocytes to control vessel diameter and blood flow [24,37]. Based on these results, we speculate that neuronal Vps35-KO may impair neuronal dendrite/axon-BV contacts, thus affecting BV arteriole contraction and diameter. Additionally, BV growth and maturation in the CNS occur at the same time as different neural cell types are generated and circuits are established [4,20,83]. We thus speculate that there may be a temporal association between neuron–terminal differentiation and BV maturation.

Our results show that neuronal Vps35-KO reduced the microglia–BV association. Microglia are specialized macrophages of the CNS involved in immune regulation, tissue development, homeostasis, and wound repair [41,70]. Microglial development starts at E8.5 and remains active, remodeling during the postnatal age [40,84,85]. It is known that microglial progenitors migrate into the brain around E8.5 and E14.5, start to colonize in the brain after E14.5, and then increase their proliferation and density gradually [86,87,88]. In addition to immune function, it is worth noting that microglia are often associated with newly forming blood vessels, where they could contribute positively to angiogenesis [89], and microglial depletion in the developing CNS results in a sparser network [90,91,92]. Similar to the microglia for the angiogenesis in the brain, tumor-associated macrophages are also known to be critical for tumor-associated angiogenesis [40,93]. These results suggest that neuronal Vps35-KO may reduce the microglia–BV association, thus decreasing BV branching.

Besides microglia, we also observed increased GFAP^+^ reactive astrocytes associated with BVs. Astrocytes are known to play multiple regulatory roles in the development and function of BV and the BBB [21]. Reactive astrocytes are strongly induced by CNS injury and disease [94,95]. Impaired BV maturation in mutant mice could cause a leaky BBB, which also activates BV-associated astrocytes [96,97]. Reactive astrocytes undergo a dramatic transformation and upregulate many genes, and functions of reactive astrocytes have been the subject of some debate [98,99,100]. Some research shows that one of the types of reactive astrocytes lacks many normal astrocyte functions, such as decreased synaptic functions and decreased phagocytic capacity. Reactive astrocytes also induce neuronal toxicity [101,102]. In addition, loss of Vps35 in developing pyramidal neurons results in not only dendritic morphogenesis defects but also neurodegenerative pathology, including glial activation [51]. We thus speculated that neuronal Vps35-KO increases GFAP^+^ reactive astrocytes and BV death and decreases BV branching.

BVs provide necessary oxygen and nutrients for neuron survival. Malformation or dysfunction of BVs in the brain results in altered blood flow and impaired neuronal function, which is implicated in multiple neurodegenerative pathologies [12,13,103]. Neurodegenerative diseases are always accompanied by a significant decrease in microvessel density and BBB pathology [6,7,12,104]. Capillary abnormalities elicit the delivery and transport of essential nutrients to neuroglial cells, and cerebral hypoperfusion can induce anatomic, physiological, behavioral, and metabolic events, such as neuronal damage, glial reactivity, visuospatial memory deficits, and cognitive disorders [29,104,105,106]. Overall, our results demonstrate a critical role of neuronal Vps35 in BV development, and impaired BV development in *Vps35^Neurod6^* mice may contribute to neurodegenerative pathology. These findings reveal the function of neuronal Vps35 in promoting BV development and maturation and contribute further insights into its functional involvement in neurodegenerative diseases.

## Figures and Tables

**Figure 1 biomedicines-10-01653-f001:**
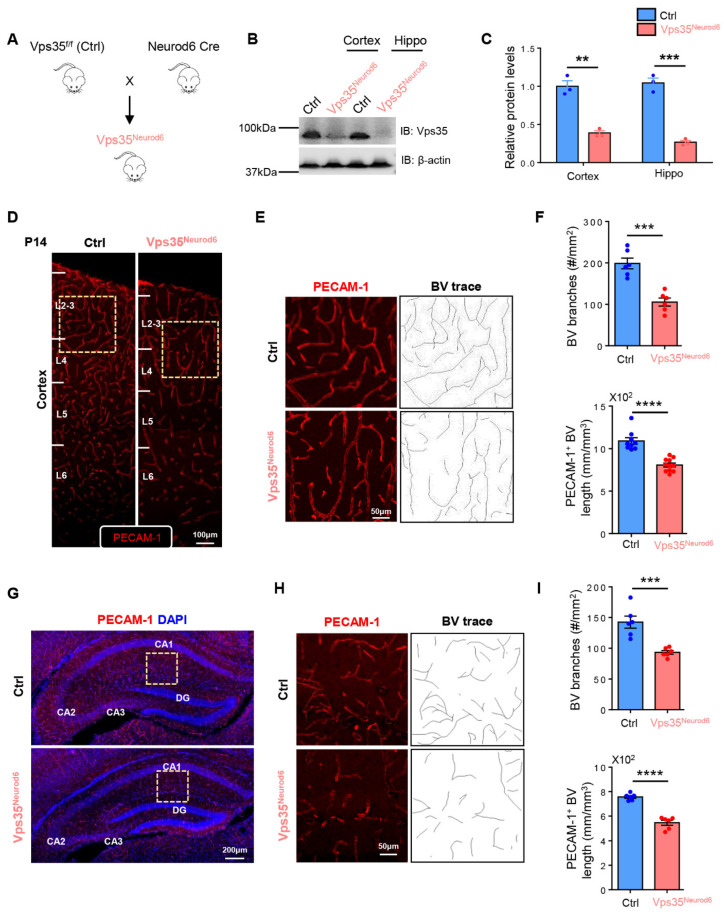
Reduced PECAM-1^+^ vessels in Vps35^Neurod6^ cortex and hippocampus. (**A**) Vps35^f/f^ mice were crossed with Neurod6 Cre mice to generate Vps35^Neurod6^ mutant mice. (**B**) Western blot analysis of Vps35 level in the cortex and hippocampus taken from Vps35^f/f^ and Vps35^Neurod6^ animals. β-actin was employed as a loading control. (**C**) Quantification analysis of relative Vps35 protein expression level from B (n = 3 animals per genotype; two-tailed unpaired *t* test). (**D**) Representative images of blood vessels (BVs) in the cortex of control and Vps35^Neurod6^ mice at age of P14. Brain sections were subjected to immunostaining analysis using PECAM-1 antibodies (red). (**E**) Higher-magnification images of the boxed regions of D and BV. (**F**) Quantification of PECAM-1^+^ vessel length and BV branches in E (n = 3 mice per genotype; two-tailed unpaired *t* test). (**G**) Representative images of blood vessels (BVs) in the hippocampus of control and Vps35^Neurod6^ mice at an age of P14. Brain sections were subjected to immunostaining analysis using PECAM-1 antibodies (red); the nuclei were stained with DAPI (blue). (**H**) Higher-magnification images of the boxed regions of G and BV. (**I**) Quantification of PECAM-1^+^ vessel length and BV branches in H (n = 3 mice per genotype; two-tailed unpaired *t* test). Scale bars as indicated in each panel. Individual data points are shown as dots with group mean ± SEM; ** *p* < 0.01; *** *p* < 0.001; **** *p* < 0.0001.

**Figure 2 biomedicines-10-01653-f002:**
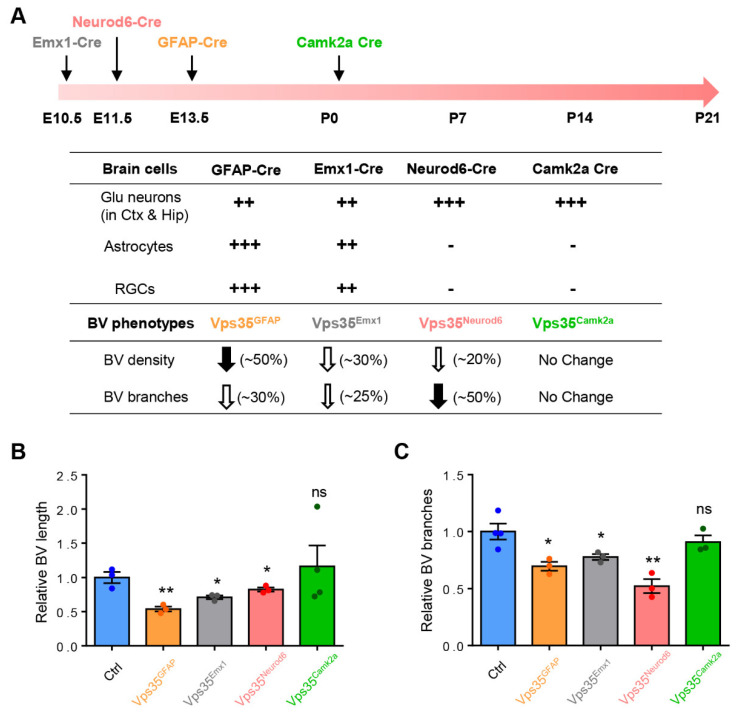
Quantification of BV length and branches of different Cre lines. (**A**) Time graph illustrating the efficiency time and space of the Emx1-Cre, Neurod6-Cre, GFAP-Cre, and Camk2a-Cre lines that were used to delete Vps35 in the embryonic and perinatal NSCs and NPCs, as well as BV phenotypes in corresponding conditional knockout mice with different Cre lines. (**B**) Summary of relative BV length in conditional knockout mice with different Cre lines (n = 3 mice per genotype; two-tailed unpaired *t* test). (**C**) Summary of relative BV branches in conditional knockout mice with different Cre lines (n = 3 mice per genotype; two-tailed unpaired t test). Individual data points are shown as dots with group mean ± SEM; * *p* < 0.05; ** *p* < 0.01; ns., not significant. ++, +++: The different amount of Cre expression in indicated cell types.

**Figure 3 biomedicines-10-01653-f003:**
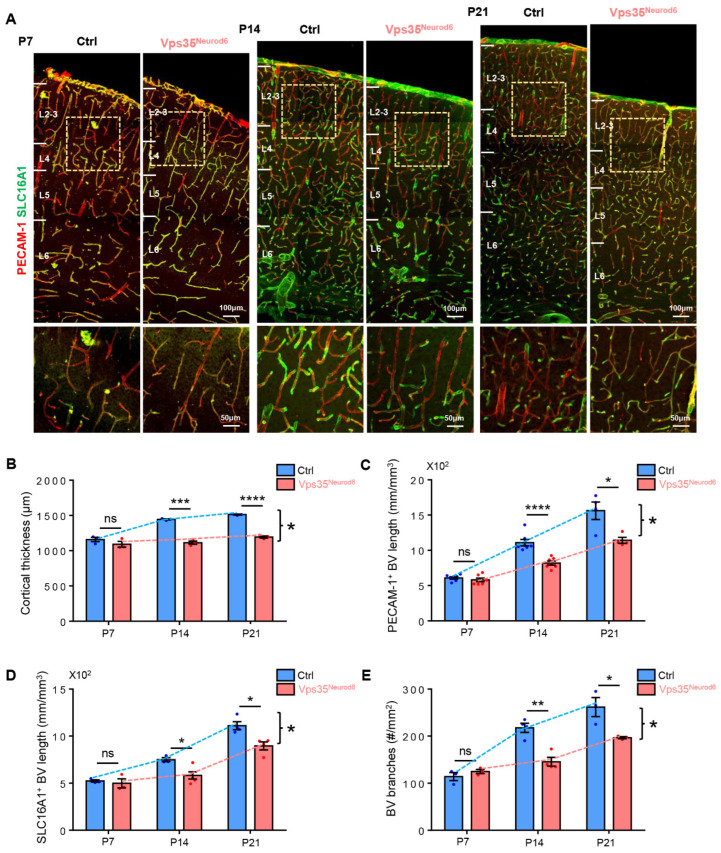
Reduced PECAM-1^+^ and SLC16A1^+^ vessels in *Vps35^Neurod6^* cortex at P14 and P21. (**A**) Representative images of immunostaining analysis using indicated antibodies in P7, P14, and P21 neocortical sections from control and *Vps35^Neurod6^* mice. Higher-magnification images of the boxed regions are shown in lower panels. (**B**) Quantification analysis revealed an age-dependent reduction in cortical thickness of *Vps35^Neurod6^* mice at P14/P21 (n = 3~4 mice per group; two-tailed unpaired t test). (**C**) Quantification of PECAM-1^+^ vessel length at indicated age (n = 3~4 mice per group; two-tailed unpaired t test). (**D**) Quantification of SLC16A1^+^ vessel length at indicated age (n = 3~4 mice per group; two-tailed unpaired *t* test). (**E**) Quantification of BV branches at indicated age (n = 3~4 mice per group; two-tailed unpaired *t* test). Scale bars as indicated in each panel. Individual data points are shown as dots with group mean ± SEM; * *p* < 0.05; ** *p* < 0.01; *** *p* < 0.001; **** *p* < 0.0001; ns., not significant.

**Figure 4 biomedicines-10-01653-f004:**
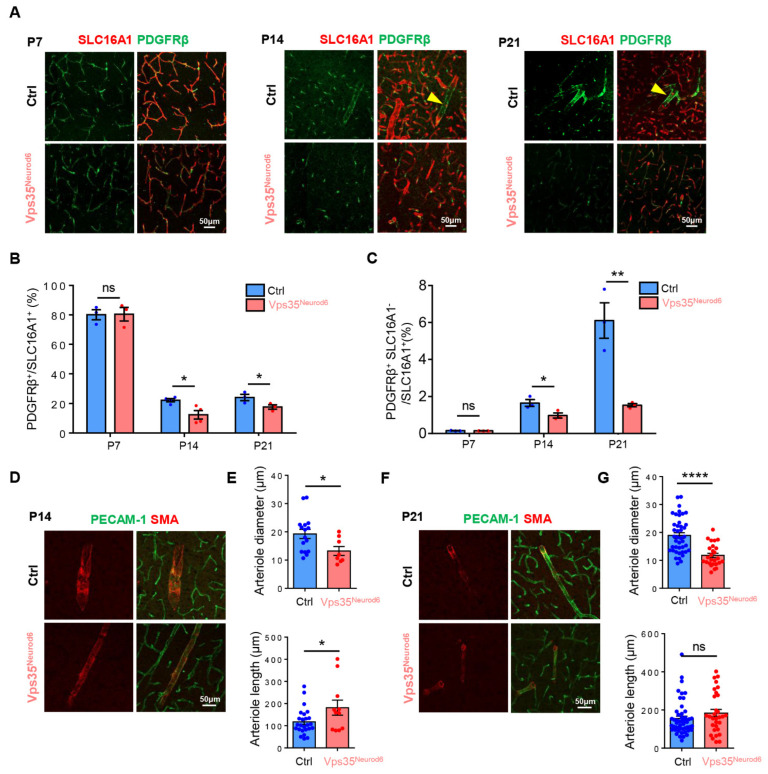
Reduced pericytes and thinner arterioles in *Vps35^Neurod6^* cortex at P14 and P21. (**A**) Representative images of cortical brain sections coimmunostained with PDGFRβ (green) and SLC16A1 (red) at indicated age. (**B**) Quantification analysis of PDGFRβ in control and *Vps35^Neurod6^* mice revealed an age-dependent decrease (n = 3~4 mice per group; two-tailed unpaired *t* test). (**C**) Quantification analysis of PDGFRβ^+^ SLC16A1^-^ cells in control and *Vps35^Neurod6^* mice revealed an age-dependent decrease (n = 3~4 mice per group; two-tailed unpaired t test). (**D**) Representative images of cortical brain sections coimmunostained with PECAM-1 (green) and SMA (marker of arterioles) (red) at P14. (**E**) Quantification of diameter and length of arterioles of control and *Vps35^Neurod6^* mice (n = 3~4 mice per group; two-tailed unpaired *t* test). (**F**) Representative images of cortical brain sections coimmunostained with PECAM-1 (green) and SMA (red) at P21. (**G**) Quantification of diameter and length of arterioles of control and *Vps35^Neurod6^* mice (n = 3~4 mice per group; two-tailed unpaired *t* test). Scale bars as indicated in each panel. Individual data points are shown as dots with group mean ± SEM; * *p* < 0.05; ** *p* < 0.01; **** *p* < 0.0001; ns., not significant.

**Figure 5 biomedicines-10-01653-f005:**
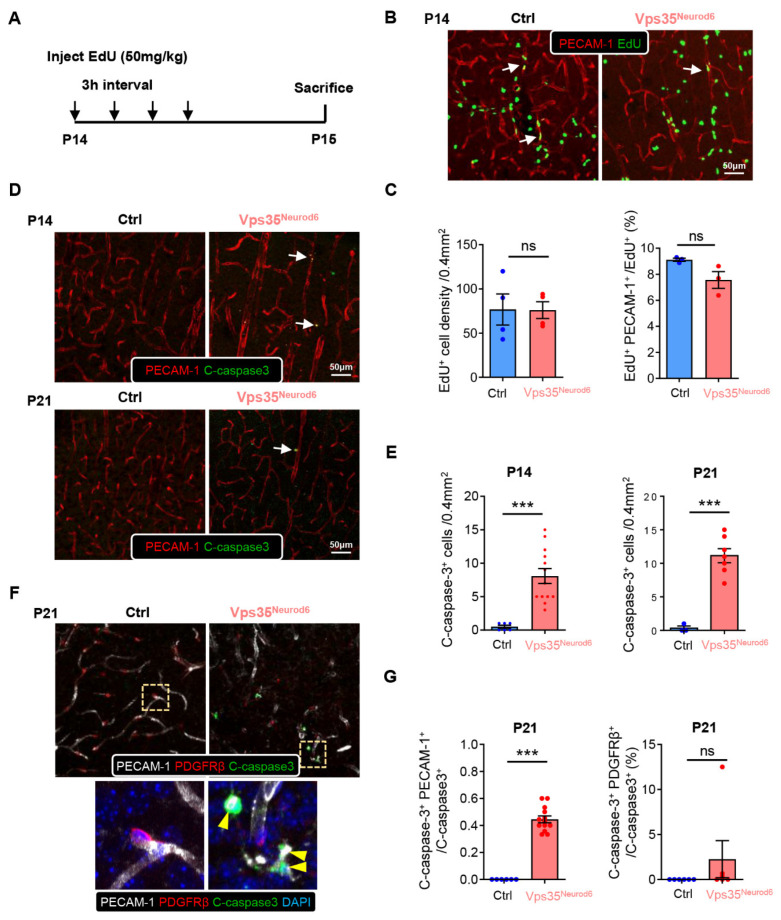
Comparable EC proliferation and increased apoptosis in *Vps35^Neurod6^* cortex at P14 and P21. (**A**) Schematic illustration of EdU injection protocol in control and *Vps35^Neurod6^* mice. (**B**) Representative images of cortical brain sections coimmunostained with EdU (green) and PECAM-1 (red). (**C**) Quantification analyses of data in B (n = 3~4 mice per group; two-tailed unpaired *t* test). (**D**) Representative images of cortical brain sections coimmunostained with cleaved caspase3 (green) and PECAM-1 (red) at indicated age. (**E**) Quantification analysis of cleaved caspase3^+^ cells that showed an age-dependent increase in apoptotic cells in the *Vps35^Neurod6^* neocortex at indicated ages (n = 3~4 mice per group; two-tailed unpaired *t* test). (**F**) Representative images of cortical brain sections coimmunostained with cleaved caspase3 (green), PECAM-1 (white), and PDGFRβ (red) at P21; the nuclei were stained with DAPI (blue). Higher-magnification images of the boxed regions are shown in the lower panels. (**G**) Quantification analysis of cleaved caspase3^+^ endothelial cells and PDGFRβ^+^ pericytes that showed an increase in apoptotic endothelial cells in the *Vps35^Neurod6^* neocortex at P21 but not in pericytes (n = 3~4 mice per group; two-tailed unpaired *t* test). Scale bars as indicated in each panel. Individual data points are shown as dots with group mean ± SEM; *** *p* < 0.001; ns., not significant.

**Figure 6 biomedicines-10-01653-f006:**
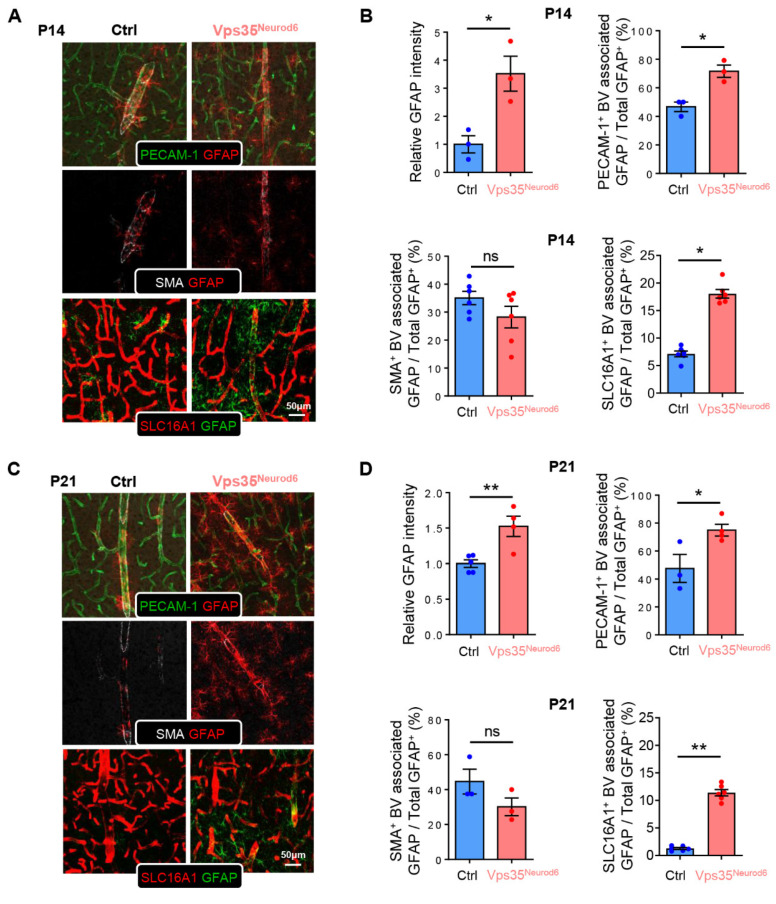
Altered GFAP^+^ astrocyte distribution in *Vps35^Neurod6^* cortex. (**A**) Representative images of cortical brain sections coimmunostained with PECAM-1, SMA, SLC16A1, and GFAP at P14. (**B**) Quantification analysis of GFAP intensity, PECAM-1^+^ BV associated with GFAP, SMA^+^ BV associated with GFAP, and SLC16A1^+^ BV associated with GFAP in control and *Vps35^Neurod6^* mice (n = 3~4 mice per group; two-tailed unpaired *t* test). (**C**) Representative images of cortical brain sections coimmunostained with PECAM-1, SMA, SLC16A1, and GFAP at P21. (**D**) Quantification analysis of GFAP intensity, PECAM-1^+^ BV associated with GFAP, SMA^+^ BV associated with GFAP, and SLC16A1^+^ BV associated with GFAP in control and *Vps35^Neurod6^* mice (n = 3~4 mice per group; two-tailed unpaired *t* test). (**E**,**F**) Western blot analyses (**E**) and quantification analysis (**F**) of GFAP level in the cortex taken from *Vps35^f/f^* and *Vps35^Neurod6^* animals. β-actin was employed as a loading control (n = 3 animals per genotype; two-tailed unpaired *t* test). (**G**) Analysis of *GFAP* mRNA level in Vps35^f/f^ and *Vps35^Neurod6^* animals at P21 (n = 3 animals per genotype; two-tailed unpaired *t* test). Scale bars as indicated in each panel. Individual data points are shown as dots with group mean ± SEM; * *p* < 0.05; ** *p* < 0.01; ns., not significant.

**Figure 7 biomedicines-10-01653-f007:**
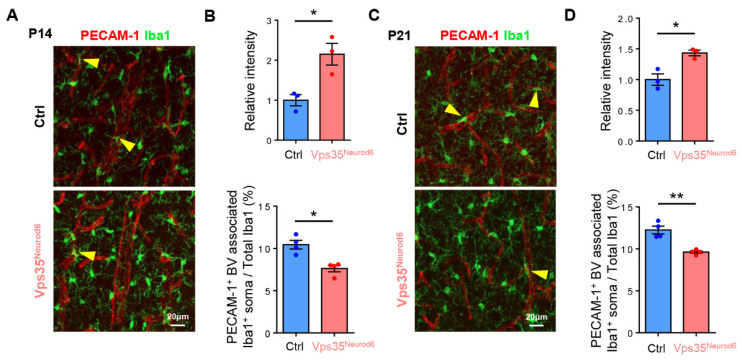
Altered Iba1^+^ microglia distribution in *Vps35^Neurod6^* cortex. (**A**) Representative images of cortical brain sections coimmunostained with PECAM-1 and Iba1 at P14. (**B**) Quantification analysis of Iba1 intensity and PECAM-1^+^ BV-associated Iba1^+^ soma of control and *Vps35^Neurod6^* mice (n = 3~4 mice per group; two-tailed unpaired *t* test). (**C**) Representative images of cortical brain sections coimmunostained with PECAM-1 and Iba1 at P21. (**D**) Quantification analysis of Iba1 intensity and PECAM-1^+^ BV-associated Iba1^+^ soma of control and *Vps35^Neurod6^* mice (n = 3~4 mice per group; two-tailed unpaired *t* test). Scale bars as indicated in each panel. Individual data points are shown as dots with group mean ± SEM; * *p* < 0.05; ** *p* < 0.01; ns., not significant.

**Figure 8 biomedicines-10-01653-f008:**
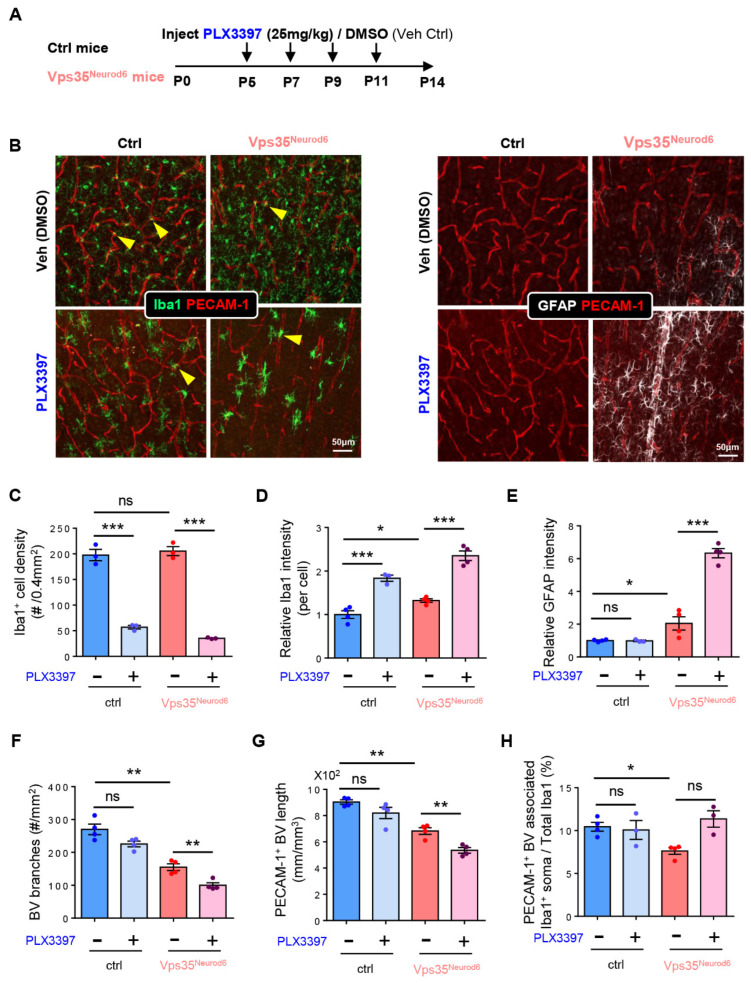
Decrease in PECAM-1^+^ vessels and increase in GFAP^+^ astrocyte in *Vps35^Neurod6^* mice depleting microglia. (**A**) Schematic illustration of PLX3397 injection protocol in control and *Vps35^Neurod6^* mice. (**B**) Representative images of cortical brain sections coimmunostained with PECAM-1 and Iba1 of control and *Vps35^Neurod6^* mice injected with PLX3397 and DMSO, respectively, at P14. (**C**) Quantification analysis of Iba1^+^ cell density in control and *Vps35^Neurod6^* mice (n = 3~4 mice per group; two-tailed unpaired t test). (**D**) Quantification analysis of Iba1^+^ cell intensity in control and *Vps35^Neurod6^* mice (n = 3~4 mice per group; two-tailed unpaired *t* test). (**E**) Quantification analysis of GFAP intensity in control and *Vps35^Neurod6^* mice (n = 3~4 mice per group; two-tailed unpaired t test). (**F**) Quantification analysis of BV branches in control and *Vps35^Neurod6^* mice (n = 3~4 mice per group; two-tailed unpaired *t* test). (**G**) Quantification analysis of PECAM-1^+^ BV length in control and *Vps35^Neurod6^* mice (n = 3~4 mice per group; two-tailed unpaired *t* test). (**H**) Quantification analysis of PECAM-1^+^ BV-associated Iba1^+^ soma in control and *Vps35^Neurod6^* mice (n = 3~4 mice per group; two-tailed unpaired *t* test). Scale bars as indicated in each panel. Individual data points are shown as dots with group mean ± S.E.M; * *p* < 0.05; ** *p* < 0.01; *** *p* < 0.001; ns., not significant.

## Data Availability

Data is contained within the article and Appendix A.

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
