# Peer review of "Critical Role of Neuronal Vps35 in Blood Vessel Branching and Maturation in Developing Mouse Brain"

_biomedicines, 2022, doi:10.3390/biomedicines10071653_

Round 1

Reviewer 1 Report

The manuscript by Zhao et al., provide a new vision of the role of neuronal Vps35 in blood vessel branching and maturation in developing mouse brain. But major revision will be indicated. The manuscript is well written, and the English used is correct enough. The study is interested in the field, but detailed need to be added.

1.- They provide evidence for embryonic neuronal Vps35 to be critical for BV branching and maturation in developing mouse brain. A decrease in BV branching and density arteriole diameter occurs in mouse embryonic knocking-out Vps35 in their experiments. They do not discuss enough what they think occurs like this. They need to add more bibliography to explain it.

2.- Why microglia were involved in embryonic situation if microglia is development in postnatal age. Authors need to explain enough that with inclusion of new bibliography too.

3.- This referee can understand that an increase in BV-associated reactive astrocytes, but they did not explain enough why, and they did not corroborate the data with other investigators. Please tell us more about it. You can do a hypothesis about it.

4.- Western-blot presented in this manuscript will be better with other images better to define the expression of proteins.

5.- Microglia have a few presences in the nervous system and these cells are involved in inflammation and defence of the organism, do you think deletion of microglia by PLX3397 enhances the deficits of BV in the mutant mice. The results are clear but is inexplicably and is difficult to believe. Microglia are not important in the formation of any BV in the nervous system.

6.- They indicate that developing pyramidal neuron Vps35’s function in regulating BV branching and maturation in developing mouse cortex and hippocampus because pyramidal neuron specific Vp35 KO cortex and hippocampus exhibited less veins and capillaries, smaller arterioles, less pericytes and impaired vascular basement membranes, but they do not present any test in that mouse to demonstrate the conclusions. This referee thinks that tests in behaviour will be necessary to demonstrate changes in the KO mice.

7.- The relation with an increase in BV-associated reactive astrocytes will be presented with a Western blot with GFAP and with RT-PCR. I think only with western blot is not enough.

8.- They suggest that the results uncover the role of pyramidal neuron Vps35 in regulating BV development by regulating the NVU, and BV-associated microglia. But what happen with astrocytes. These kinds of cells can regulate the NVU and BV associated microglia more efficiently that neurons in any place of nervous system. There are more astrocytes than neurons in the nervous system and microglia are not a neural cell, so why author think neurons are the only cells to control all the system? Explain their hypothesis with results of other authors.   

I think authors have a great experience in nervous system, only they must be more accurate in their conclusions.

Reviewer 2 Report

The present manuscript was well organized and interest to readers.

There were some spelling errors and they should be corrected at the stage of proof.

Round 2

Reviewer 1 Report

Accept